# Galectin-1 Used in Assisted Reproduction—Embryo Safety and Toxicology Studies

**DOI:** 10.3390/molecules28020859

**Published:** 2023-01-15

**Authors:** Marcelo Roncoletta, Nathali Adrielli Agassi de Sales, Fernando Sebastian Baldi Rey, Guilherme Camargo Ferraz, Erika da Silva Carvalho Morani

**Affiliations:** 1Yoni Group, Inprenha Biotecnologia, Jaboticabal 14897-899, São Paulo, Brazil; 2Laboratório de Fisiologia do Exercício e Farmacologia (LAFEQ), Departamento de Morfologia e Fisiologia Animal, Universidade Estadual Paulista (UNESP), Jaboticabal 14884-900, São Paulo, Brazil; 3Departamento de Zootecnia, Universidade Estadual Paulista (UNESP), Jaboticabal 14884-900, São Paulo, Brazil

**Keywords:** reproductive safety toxicology, Tolerana^®^, galectin-1, recombinant protein

## Abstract

Galectin-1 has been cited as a mediator involved in preventing early embryonic death in mammals and is implicated in maternal–fetal tolerance. Galectin-1 is also a reasonable tool to improve fertility in assisted reproduction procedures. As recommended in the ICH guidelines (S5-R2 and S6-R1) and based on bioethical concerns, we chose bovine embryos (BE) to assess in vitro embryo development as part of a larger reproductive safety and toxicology study in progress. The design considered in vitro embryo development using rHGAL-1 supplementations (in three different concentrations) of the in vitro embryo culture (IVP) media. Based on procedures for the commercial in vitro production of BE using oocytes aspirated from slaughterhouse ovaries, rHGAL-1 supplementation was performed in two experiments: In Experiment 1 on oocyte maturation, involving IVM medium supplementation, and in Experiment 2 on culture step IVC, involving supplementation with an SOF medium. IVP commercial procedures were used, with three IVP replicates per experiment, and the oocytes we distributed into four groups of treatment (one control group and three different dosages of rHGAL-1 to supplement both IVM and SOF media using 2, 20, and 40 µg·mL^−1^, respectively. A total of 967 (Experiment 1) and 1213 (Experiment 2) oocytes were aspirated and submitted to the IVP procedure. There was no damage to the in vitro bovine embryo growth when considering cleavage percentage (%CLE), blastocyst development (Bl, Bx, Bh, and B) at Days 7 and 8, or an amount of rHGAL-1 supplementation ≤20 µg·mL^−1^. The immunohistochemistry assay with D8 embryos cultivated using rHGAL-1 supplementation on the culture medium (SOF medium) demonstrated the presence of exogenous GAL-1 distributed in mass cell and trophoblastic cells, and the profile observed was dependent on exogenous supplementation, which was most evident in hatched embryos. The findings confirmed the use of a reasonable amount of rHGAL-1 for in vitro embryonic development and would make the use of rHGAL-1 in assisted reproduction in humans more reliable and safer. Even though it was not the objective of the study, we verified that supplementation with 2 µg·mL^−1^ significantly improved some of the evaluated parameters of embryonic development (%BlD7, %BD7, %BlD8, %BhD8, and %BD8).

## 1. Introduction

Galectin-1 (GAL-1) has been cited as a mediator involved in preventing early embryonic death in mammals and in maternal–fetal tolerance (both innate and adaptive) and is associated with regulation and modulation of immunological responses and adherence to the endometrium. GAL-1 also contributes to placentation, controlling the development, migration, and trophoblastic invasion essential in early gestational development [1,2,3,4,5,6]. GAL-1 is a 14 kDa lectin with a high affinity to carbohydrates (beta-galactoside and lactose) and is expressed in different tissues but with a considerable amount of mRNA and protein expression in the endometrium and placenta (in trophoblasts, stromal cells, villous endothelium, syncytiotrophoblast apical membrane, villous stroma, maternal decidua, and fetal membranes), as well as in fetal membranes (amnion, chorion–amnion mesenchyma, and chorion) of different mammal species, including humans.

GAL-1 can be detected in the endoplasm of mouse oocytes and preimplantation embryos at all stages, including the zygotes, two-cell embryos, four-cell embryos, eight-cell embryos, and blastocysts (mainly in trophectoderm cells) [7]. Gal-1 expression was observed in 3–5-day-old human embryos, potentially increasing trophoblast attachment to the uterine epithelium [3,8]. Gal-1 is an important contributor to fetal–maternal tolerance and has been extensively described and reviewed [1,3]. Several immune cells with essential roles in the establishment and maintenance of pregnancy synthesize and respond to GAL-1, e.g., CD4+ CD25+ regulatory T-cells, which play a vital role in tolerating immunogenic paternal alloantigens [1,9,10], and the regulation of the expression of human leukocyte antigen (HLA-G) in extravillous trophoblasts, contributing to tolerance via its interaction with immune and trophoblast cells [8].

Our research group recently evaluated the effect of a single intrauterine dose of human GAL-1 buffered based on the pregnancy rate in inseminated cows. The findings suggested that one dose of eGAL-1 was reasonable in a beef cattle artificial insemination routine and considerably improved the pregnancy rate [11,12]. To explore the future possibilities of using exogenous rHGAL-1 administration in human reproduction procedures (artificial insemination, embryo transfer, etc.), considerable efforts are being made to understand the relevant pharmacokinetics, toxicology, and pharmacological security, all based on *ICH Topic S5-R2* (1994) [13] and *ICH Guideline S6R1* (2011) [14]; the present article is part of this corpus. This paper complements reproductive safety toxicology studies and analyzes embryo development after being supplemented with media from an in vitro embryo culture (IVP) with rHGAL-1 in three different dosages. Herein, we selected bovine embryos due to their availability and bioethical concerns, which enabled us to design a study with a very reasonable number of oocytes and embryos. The complete reproductive safety toxicology study also includes a histopathological evaluation of representative organs of necropsied rat fetuses (on the 7th and 14th day after mating and following the 21st day after birth), considering intrauterine administration of rHGAL-1 (using a surgical method). This study is still evaluating its results. The goal of this article is to demonstrate the safe amount of protein to use in human-assisted reproduction procedures with pharmacological safety for the embryos. To evaluate pharmacological security for the mother, other nonclinical trials were carried out.

## 2. Results

The results obtained for embryonic development (n = 967 aspirated oocytes) with the IVP procedure plus supplementation of the IVM medium (Experiment 1) are shown in Table 1, which details the IVF parameters (%CLED2) observed 36–48 h after IVF, obtained using the three replicates performed according to the design of Experiment 1.

On average, 79% of CLED2 presented no statistical differences between groups (*p* < 0.05) when the IVM medium was supplemented with the test item (eGAL-1). The statistical analysis model used compared the results of the embryonic development parameters based on the groups of treatment (G1, G2, G3, and G4). Table 1 illustrates these results. Here, %CLED2 represents 73.88 ± 1.01% (G1), 82.02 ± 0.83% (G2,) 70.69 ± 1.06% (G3), and 79.43 ± 0.89% (G4). There were no statistically significant differences (*p* < 0.05) between the treatment groups for the cited parameters. Notably, one replicate of G3 had microbiological contamination and was thus excluded from the statistical analysis.

Table 1 also details the embryonic development parameters observed on D7 and D8. Like %BD, %Bl, %Bx, and %Bh, all observations were taken seven (D7) and eight (D8) days after the IVC stage of the procedure. Considering all D7 results obtained, we observed statistical differences between %BlD7 with G2 (20.32 ± 0.32) and G4 (11.40 ± 0.25, *p* < 0.06), between %BD7 with G2 (34.07 ± 0.53) and G3 (18.30 ± 0.4, *p* < 0.05), and between G2 and G4 (17.42 ± 0.4, with *p* < 0.05). Considering %CLED2, %BxD7, and %BhD7, they did not present any statistical differences according to the level of eGAL-1 supplementation. Considering all D8 results obtained, we observed statistical differences between %BlD8 with G2 (10.48 ± 0.15) and G3 (6.50 ± 0.12, *p* < 0.078), between G2 and G4 (6.3 ± 0.14, *p* < 0.1), between %BhD8 with G2 (15.25 ± 0.27) and G1 (7.78 ± 0.02), between G2 and G3 (6.65 ± 0.18, *p* < 0.03), with %BD8 between G2 (37.69 ± 0.56) and G3 (23.28 ± 0.48, *p <* 0.09), and between G2 and G4 (*p* < 0.05). Considering %BxD8, it did not present any statistical differences according to the level of eGAL-1 supplementation.

The results obtained in the second experiment, with rHGAL-1 supplementation in the SOF medium, are described in Table 2. The results obtained for embryonic development (n = 1213 aspirated oocytes equally distributed into the four groups) were 79.41% of CLED2 (on average), with statistical differences between some groups: %CLED2 between G1 (85.04 ± 1.99), G3 (78.21 ± 2.31), and G4 (72.64 ± 2.50), both with *p* < 0.05; and between G2 (81.42 ± 2.17) and G4 (*p* < 0.05). Note that no supplementation was performed until this stage of the procedure.

Figure 1 illustrates the quality of the embryos developed in each treatment group (G1, G2, G3, and G4). No evidence of morphological damage was observed after IVM medium supplementation with rHGAL-1.

Table 2 details the other embryonic development parameters observed on D7 and D8, after SOF supplementation with rHGAL-1, including parameters such as %BD, %Bl, %Bx, and %Bh, all observed seven (D7) and eight (D8) days after the IVC stage of the procedure. Considering all D7 results obtained, we observed statistical differences between %BlD7 with G3 (27.40 ± 4.41) and G4 (16.22 ± 2.95, *p* < 0.068), between G1 (12.98 ± 0.25) and G3 (5.51 ± 0.23, *p* < 0.039), between G1 and G4 (4.15 ± 0.3, *p <* 0.01), and between G2 (11.72 ± 0.25), G3 (*p <* 0.06), and G4 (*p <* 0.01), all considering %BxD7. For %BD7 (the summary of all embryo stages), the statistical analyses showed significant differences between G1 and G4 (33.53 ± 2.59 vs. 22.56 ± 2.28, respectively; *p <* 0.01) and between G2 and G4 (34.67 ± 2.61 vs. 22.56 ± 2.28, respectively; *p <* 0.01).

Finally, considering the D8 results obtained, we observed statistical differences in %BxD8 between G1 and G4 (20.74 ± 1.45 vs. 10.74 ± 1.10, *p <* 0.001), G2 and G4 (19.13 ± 1.40 vs. 10.74 ± 1.10, *p <* 0.002), and G3 and G4 (17.81 ± 1.67 vs. 10.74 ± 1.10, *p <* 0.008). For %BhD8, there were differences between G1 and G2 (10.01 ± 1.20 vs. 13.60 ± 1.37, *p <* 0.09) and in G2 to G4 (13.60 ± 1.37 vs. 8.77 ± 1.12, *p <* 0.03). For %BD8, we observed differences between G1 and G4 (36.73 ± 0.52 vs. 24.14 ± 0.48, *p <* 0.004), in G2 to G4 (36.62 ± 0.56 vs. 8.77 ± 1.12, *p <* 0.005), and between G3 and G4 (34.74 ± 0.48 vs. 24.14 ± 0.48, *p <* 0.01).

Figure 2 illustrates the quality of the in vitro bovine embryos developed in each treatment Group (G1, G2, G3, and G4). No morphological damage was observed with rHGAL-1 supplementation of the SOF medium.

The immunohistochemistry assay with embryos cultivated using rHGAL-1 supplementation on the culture medium (SOF medium) indicated the presence of exogenous GAL-1 distributed in mass cells and trophoblastic cells. Figure 3, Figure 4, Figure 5 and Figure 6 illustrate the presence of yellow–dark spots detected after DAB staining and their relationship with the amount of rHGAL-1 used in the medium supplementation (G2, G3, and G4). Here, more protein corresponded to more detectable dark spots. Exogenous GAL-1 penetrated the zona pellucida (as shown in blastocysts with an intact zona pellucida) but was more evident in the detection of spots in the trophoblastic cells of hatched embryos. Another observation on the immunohistochemistry assay was the possibility of detecting yellow–dark spots, even in a small amount, which was confirmed in embryos of the control group without medium supplementation. (Figure 3).

## 3. Discussion

This study aimed to complement current pharmacological knowledge, specifically the reproductive safety toxicology of rHGAL-1 as an active ingredient in current and potential products intended to assist reproduction in both production animals and humans.

The hypothesis of using exogenous GAL-1 as a tool to increase fertility was supported by the present authors and has been the subject of study since 2008, with the assumption that supplementation of an adequate amount of this protein into the lumen uterus would yield the desired effect on reproduction. To understand the impact of this protein on embryonic development greatly facilitates this goal.

The scientific support for using the administration of exogenous was first proposed in [2], which demonstrated a higher rate of pregnancy loss in female knockout mice for the LGALS1 gene, the gene responsible for the expression of GAL-1, along with other theoretical and practical research demonstrating the intrinsic participation of GAL-1 in important physiological events related to the maternal recognition of pregnancy, either based on immunological or biological aspects such as embryonic elongation and adhesion, trophoblastic development, and placentation [1,2,3,4,5,10,15,16].

There are many particularities between the placentation and embryo elongation of different mammal species. However, the authors in [10,16,17] noted that GAL-1 has a high degree of structural conservation, dimerization, and binding properties with carbohydrates and integrins (adhesion proteins), suggesting that these properties are conserved among vertebrates and maintain a pattern of gene expression among the different types of the placenta (deciduous or not), supporting the decision to use a human gene to produce the recombinant proteins for other species. Previous studies have demonstrated the biological functionality of rHGAL-1 in improving pregnancy rates in cows, suggesting its potential use in other species [11,12].

As part of the non-clinical studies needed to start clinical trials in humans (i.e., phase 1 onwards, for which the goal is to use intra-uterine rHGAL-1 dose administration, a per-artificial-insemination procedure, or a per-embryo transfer procedure to improve the success rate of pregnancy), we published pharmacokinetics study results [18]. As a complement to this past work, the present study explores the pharmacological aspects (reproductive safety toxicology) of rHGAL-1 tested using three different dosages as supplementation for IVM and SOF media during IVP procedures. The choice of bovine in vitro embryos was based on bioethical questions and the potential to test a larger number of oocytes and embryos available for IVP procedures. We used bovine ovaries originating from slaughterhouse ovaries. Additionally, to follow the recommendations of the ICH guides, such as those for the preclinical safety evaluation of biotechnology-derived pharmaceuticals [13,14], as is the case presented here, we used a recombinant protein-based product.

The suggested form of administration for exogenous rHGAL-1 in assisted reproduction procedures involves administration of the protein solution directly into the uterine lumen during the insemination act or embryo inovulation using an equivalent dose of 8–16 µg·mL^−1^ rHGAL-1 (supposing a uterus lumen volume of 50 to 25 mL [19]). Hence, the present study helps explain the effects of exogenous protein presence and contact on embryonic development, even though during artificial insemination procedures where rHGAL-1 is used on the day of insemination, the protein and embryo will not come into contact. According to the authors in [18], after 48 h of uterus administration, in rats, no more rHGAL-1 was detected on the uterus tissue homogenate.

Even considering the particularities of embryonic development in vivo and in vitro and the amount of protein used as a supplement, we verified the absence of harmful effects when comparing G2 (2 µg·mL^−1^) with G1 (control group) according to most parameters, including %CLED2, %BlD7, %BxD7, %BhD7, %BlD8, %BxD8, and %BD8, as evaluated in Experiment 1 (IVM medium supplementation), and %BlD7, %BxD7, %BD7, %BlD8, %BxD8, and %BD8 in Experiment 2 (SOF medium supplementation). Only %Bh on D7 and D8 demonstrated improved percentage ratios when using 2 µg·mL^−1^ supplementation compared to the control group. However, most of the other embryonic development parameters (%BlD7, %BhD7, %BD7, %BlD8, %BxD8, %BhD8, and BD8), decreased considerably when a higher amount of rHGAL-1 was added (40 µg·mL^−1,^) in both experiments. The results of 20 µg·mL^−1^ supplementation (G3) during the IVM step (Experiment 1) appeared very similar to those of G4. In Experiment 2 (SOF supplementation), the results of G3 were very similar to those of G1 and G2 but differed somewhat from those of G4 in the embryonic parameters (%BlD7, %BD7, %BxD8, %BhD8, and %BD8. This information could be used in the future to support new experiments to find an optimal dose of rHGAL-1 supplementation and thus improve the IVP results, principally in an SOF medium or IVC procedure.

%CLED2 demonstrated no statistical differences between groups after rHGAL-1 supplementation in the IVC medium. We observed some differences between groups in this parameter during Experiment 2. However, these differences were not because of the supplementation, as rHGAL-1 was added to the SOF medium only during the IVC procedure, not before it. This percentage ratio was larger in G1 than in other groups, which could interfere with the other parameters obtained for each set of IVP procedures.

The results describe the mean ± SD of each embryonic development parameter observed on D7 and D8 in the groups. Nevertheless, there is consensus that in IVP commercial protocols, it is normal to find variation between embryonic developments based on the embryonic parameters measured. Further, microbiological contamination may interfere with this issue. For this reason, on some drops of G3 in Experiment 1, this IVP replicate was excluded from the statistical analyses. However, the statistical analyses were able to concisely demonstrate the effect of supplementation and its interference with embryonic development parameters. Nevertheless, no morphological evidence (based on an optical evaluation) was observed in the embryos from different groups of treatment.

Although supplementation with an rHGAL-1 dose of ≤20 µg·mL^−1^ was demonstrated to be safe for bovine embryonic development, a dose 5 to 10× higher than the safe range for doses used in the effectiveness studies (2–4 µg·mL^−1^ of rHGAL-1 per AI procedure) would be a useful range to investigate the efficacy of an exogenous dose of rHGAL-1 on reproductive-assisted procedures for different species. For other species, the idea is to maintain the dosage proportion and look for improvements in the fertility ratio. In both experiments with IVP embryos, and considering G3 and G4, we extrapolated the amount of protein used on these when comparing with the protein amount used in the efficacy tests in cows, creating an exciting range of possible doses to test. Studies have shown that the supplementation of an in vitro oocyte maturation medium with macromolecules (including some proteins) influences in vitro development and, consequently, embryonic development, as these macromolecules interfere with the success of oocyte maturation competence, where nuclear and cytoplasmic reprogramming events occur [20]. rHGAL-1 could be considered another option for these macromolecules to improve the efficacy of IVP procedures, but more studies are needed to confirm this hypothesis.

The immunohistochemistry assay demonstrated rHGAL-1 detection in D8 embryonic structures with different levels of detection (spot intensity of detection), suggesting that, according to the amount of exogenous protein used to supplement the SOF medium (Experiment 2), more spots of immunoreaction were detected. Lgals1 expression was previously observed in 3–5-day-old human embryos, potentially increasing trophoblast attachment to the uterine epithelium [8]. The following immunohistochemistry assay results suggested that the larger amount of immunoreactive spots detected in subsequent embryonic structures (hatched and expanded) was due to complementation of the detected protein with the endogenous expression of GAL-1 (bovine). The cross-reaction between IgG anti-human GAL-1 and bovine-GAL1 is high, and the authors detected both in ELISA assays; the degree of similarity between them is greater than 89%.

This study obtained essential answers to questions intrinsic to recombinant protein-based biopharmaceuticals. Notably, more than 2000 aspirated oocytes and cultures were considered in each of the two experiments, reinforcing the accuracy of this study.

The experiments still leave an open question: Is it possible for rHGAL-1 to interact with different stages of embryonic development (especially the steps with differentiated trophoblasts), thereby altering the effectiveness of the embryo in the maternal recognition of pregnancy, or is the role of rHGAL-1 in increasing reproductive efficiency related only to the uterine endometrium? New experiments are underway to elucidate this question and explore rHGAL-1 supplementation. Another relevant experiment will analyze reproductive toxicity in female rats that received a single dose of rHGAL-1 via intrauterine administration using surgical procedures, based on the work in [13]. The goal of this study is to obtain information on different phases of pregnancy, including the first stage (conception to implantation, evaluating adult female reproductive functions and comparing the number of, as well as evaluating, the gestational sacs), the second stage (implantation to the closure of the hard palate, assessing adult female reproductive functions, embryonic development, and major organ formation), the third stage (closure of the hard palate to the end of pregnancy, evaluating mature female reproductive functions, fetal development and growth, and organ development and growth), and the fourth stage (birth to weaning, evaluating adult female reproductive functions, neonate adaptation to extrauterine life, and preweaning development and growth up to 14 days). With both experiments, we will be able to better discuss a single intrauterine rHGAL-1 dose and its interaction with embryos, as well as the consequences for embryo and fetus development and the mother’s health. Previous studies have noted that GAL-1 plays a critical role in regulating embryo implantation and maintaining early placenta functions (trophoblast differentiation, migration, and selective gene expression in placentation) [8]. GAL-1 expression increases significantly during pregnancy, and several studies indicated the potential use of Lgals1 as a biomarker for miscarriage, recurrent fetal loss, and preeclampsia (PE) [3]. The results showed decreased expression of GAL-1 in the trophoblastic cells of women with early pregnancy loss (using proteomic analysis) [15]. Another study demonstrated increased expression of GAL-1 in invasive extravillous trophoblasts in human placenta during the first trimester, which was previously shown to promote syncytium formation [8,21]. The authors suggested that the blockage of GAL-1-mediated angiogenesis or lectin can disturb the processes associated with good placentation. Data from the authors (improvements in the amount of GAL-1 in the bovine uterus) can be cited as a good example of the impact of this protein on the pregnancy rate (11, 12).

## 4. Materials and Methods

### 4.1. rHGAL-1 Production and Purification

The method for obtaining rHGAL-1 was determined via the manufacturing process of Inprenha Biotecnologia^®^ and was previously described in its entirety [11,12]. Aliquots of *E. coli* strains transformed with the vector insertion containing the GAL-1 gene (pET-29a(+)+lgals-1 gene were grown in systems with an LB broth base medium containing kanamycin sulfate until we obtained an optimal bacterial growth rate, as demonstrated by optical density. Regarding to subcloning of GALl-1 into the pET-29a(+) expression vector, the GAL-1 Consensus Coding Sequence (CCDS) CCDS13954.1 (length 408 nt) was synthesized and subcloned with the juxtaposed insertion of the desired sequence immediately after cutting the RBS ribosome binding site sequence of a pET-29a (+) expression vector in NdeI/HindIII (GenScript^®^). This construct was then used for competent transformation of the Rosetta strain of *Escherichia coli*, maintained in a cell bank. The induction of expression was achieved by adding isopropyl-D-thiogalactopyranoside (Sigma-Aldrich) to the culture. After the induced growth period, the bacterial suspension was retained via microfiltration on a hollow fiber membrane (0.22 µm, Cytiva) and centrifuged at 5000 g for 15–20 min at 4 °C, always with the supernatant being discarded. The bacterial crude recovered as a pellet was then subjected to bacterial lysis. For bacterial lysis, the crude or bacterial pellet was resuspended in phosphate saline lysis buffer (1X PBS; 136.8 mM NaCl, 2.7 mM KCl, 6.4 mM Na2HPO4, and 0.9 mM KH2PO4, pH 7.4) containing 14 mM mercaptoethanol, protease inhibitor EDTA-free, lysozyme-1, RNAse A-Type 3A, and DNAse I Type IV-10. All components were from Sigma-Aldrich. The pellet diluted in a lysis buffer (chemical lysis) was subjected to constant homogenization for 70 min and then sonicated for three cycles of 15 s each in a Vibra-Cell sonicator, Sonics (mechanical lysis), with intervals of 20 s between each cycle. The bacterial lysate was then clarified via centrifugation at 7000 g for 20 min at 4 °C and filtered through a 1.0 µm filter (Whatman) with a peristaltic pump (maximum pressure of 4 BAR). The lysate was subjected to 3 stages for protein purification via chromatography in an AKTA Protein Purification System (Cytiva) to obtain a buffered protein solution containing only GAL-1. The first step was based on affinity chromatography on agarose–lactose columns (Sigma-Aldrich), followed by size-exclusion chromatography (Sephadex G-25, Cytiva) and another round of affinity chromatography (PIERCE High-Capacity Endotoxin Removal Resin column, Thermo Scientific) for the removal of bacterial endotoxins (LPS). After all chromatographic steps, the protein concentration was determined by spectrometry (Abs 280 nm) and expressed in milligrams of protein per milliliter (mg.mL^−1^). The concentration was then subjected to sterilizing filtration (0.22 μm, PES membrane). Purified protein batches were subjected to the last stage of industrialization only if they reached compliance with the quality standards predetermined by the company, including protein concentration (1.05 ± 0.05 mg.mL^−1^), microbiological status, protein bioactivity (hemagglutination test), molecular weight analysis via SDS-PAGE, size-exclusion chromatography (SEC), protein secondary structure analysis (circular dichroism analysis), aggregate detection and molecular size determination by dynamic light scattering (DLS) analysis, and endotoxin quantification (LPS). Protein identity was confirmed using liquid chromatography–mass spectrometry (LCMS) and nucleotide sequence confirmation of human galectin-1 cDNA-galectin-1 (*Homo sapiens*) with Consensus Coding Sequence (CCDS) CCDS13954.1 (https://www.ncbi.nlm.nih.gov/projects/CCDS/CcdsBrowse.cgi?REQUEST=ALLFIELDS&DATA=CCDS13954.1&ORGANISM=0&BUILDS=CURRENTBUILDS, accessed on 12 January 2023).

### 4.2. IVP Procedure Steps

For all IVP replicates, the steps were as follows. Day −1: (a) slaughterhouse ovary collection, (b) oocyte aspiration, (c) IVM medium pre-equilibration in 5.5 ± 0.5% CO_2_ and 38.5 ± 1.0 °C atmosphere for 2 h minimum, (d) oocyte washing twice, using a wash medium, (e) transferring washed oocytes to pre-equilibrated IVM medium drops; (f) oocyte and COC evaluation, and (g) pre-selected oocyte maturation procedure; Day 0: (h) IVF medium pre-equilibration in 5.5 ± 0.5% CO_2_ and 38.5 ± 1.0 °C atmosphere for 2 h minimum, (i) washing maturated oocytes twice using a TL semen medium, (j) oocyte transfer to pre-equilibrated IVF medium drops, (k) semen preparation including centrifugation of thawed semen, sperm concentration determination, and sperm dosage determination, an (l) adding the sperm dosage to the IVF drops with prematurated oocytes; Day 1: (m) SOF medium pre-equilibration in 5.5 ± 0.5% CO_2_ and a 38.5 ± 1.0 °C atmosphere for 2 h minimum and (n) transferring zygotes to pre-equilibrated SOF medium droplets; Day 2: (o) cleavage percentage ratio determination; Days 3 and 5: (p) SOF medium replacement on the drop using a pre-equilibrated SOF medium; Day 7: (q) embryo development evaluation at D7 and (r) embryo collection for immunohistochemistry; Day 8: (s) embryo development evaluation at D8 and (t) embryo collection for immunohistochemistry.

### 4.3. Experiment 1—IVM Medium Supplementation

IVPs of bovine embryo procedures were performed considering four groups of treatments, one control group (no supplementation) and three groups with rHGAL-1 supplementation on BIOK IVM medium, with 2 µg·mL^−1^ of rHGAL-1 (Group 2), 20 µg·mL^−1^ of rHGAL-1 (Group 3), and 40 µg·mL^−1^ of rHGAL-1 (Group 4). The IVM media used for each group were prepared, excluding the water amount (QS) and adding the QS amount of volume of the protein solution (this was done to avoid exchanging the amount of the other medium components). The protein solution was added on the day of use of the IVM medium in the IVP routine. Other media used in the IVP routine were not supplemented. Three different IVP replicates were used, with equal distribution of the aspirated oocytes per group and per replicate.

### 4.4. Experiment 2—SOF Medium Supplementation

IVPs of bovine embryos were performed considering four groups of treatments, one control group (no supplementation), and three other groups with rHGAL-1 supplementation on a BIOK SOF medium with 2 µg·mL^−1^ of rHGAL-1 (Group 2), 20 µg·mL^−1^ of rHGAL-1 (Group 3), and 40 µg·mL^−1^ of rHGAL-1 (Group 4). The SOF media used for each group were prepared, excluding the water amount (QS) and adding the QS amount for the volume of protein solution (this did not change the amount of the other media components). The protein solution was added on the use day of the SOF medium in the IVP routine (used during development and feeding). Other media used in the IVP routine were not supplemented. Three different IVP replicates were used, with equal distribution of the aspirated oocytes per group and per replicate.

### 4.5. Dose Selection and Justification

The selection of rHGAL-1 doses used in the IVP experiments was based on the proportion of the product (the adequate amount already established for artificial insemination procedures in cattle, equal to 200 ± 10 µg of rHGAL-1) over the size of the uterus lumen of a bovine female (25–50 mL per horn, according to buffer volume, in mL) used for embryo flushing during the embryo transfer procedure, according to the procedure in [19], which considered a concentration of 4–2 µg·mL^−1^. We used a 100 µL in vitro oocyte environment drop that was 1000× less than the uterine lumen volume. Therefore, a 1000× lower dose in the IVP drops was used (i.e., 2 µg·mL^−1^ or 0.2 µg in 100 µL of IVP medium). This was the lowest dose employed (group 2). We considered medium doses as 10× more protein/mL in each medium (Group 3, with 20 µg·mL^−1^) and high doses as 20× more protein/mL in each medium (Group 4, with 40 µg·mL^−1^).

### 4.6. Animals

Cows sent for slaughter in commercial slaughterhouses were registered with the relevant agencies. However, breed, age, and nutrition were not recorded or controlled. The slaughterhouse decided the types of animals accepted for slaughter. Ovaries from cows slaughtered on the same day were all grouped and sent to the IVP laboratory and were considered one replicate of IVP. Both ovaries from each cow were used for oocyte aspiration. In total, 229 cows were used in both experiments (1 and 2). All ovaries were kept in an isothermal box until being manipulated in the laboratory. The distance between the slaughterhouse and the IVP laboratory was approximately 180 km.

### 4.7. IVP

All procedures for the in vitro bovine embryo culture were performed according to the manufacturer’s instructions using commercial media such as a BIOK WASH medium to wash the oocytes and remove follicular fluid and cell debris, a BIOK IVM medium to incubate the selected oocytes during the maturation procedure, and a BIOK TL medium to set up the Percoll gradient. Additionally, during the first washes of the oocytes when transferring them to the IVF step, a BIOK PERCOLL medium was used in the processing of semen after thawing; in this way, it was possible to obtain the live sperm pellet to be used in fertilization. BIOK IVF medium was used in the processing of the semen after passage through the Percoll gradient, in correcting the sperm concentration of the inseminating dose, in washing the oocytes when transferring them to the IVF plate, and in the fertilization plates. BIOK DESNUD medium was developed to help in the denudation of zygotes, as this substance facilitates the detachment of granulosa cells without the need for excessive pipetting, thereby preventing lesions in their zona pellucida. BIOK SOF medium was used for washing zygotes when transferring them to IVC plates and in embryonic development culturing. For each IVP replicate, identical medium batches were used in all treatment groups, and only one semen batch was used for all experiments.

#### 4.7.1. Oocyte Aspiration and COC Selection

After arriving at the IVP laboratory, the ovaries from the cows slaughtered in each replicate were washed with a warm physiological solution (37 °C) using a sieve repeatedly until we removed all blood. After that, the cleaned ovaries were incubated at 37 °C in a glass bottle until all ovaries were aspirated. Ovum aspiration was performed using a 30 × 10 mm^2^ needle and syringe, aspirating all visible follicles, excluding only follicles larger than 30 mm. The follicular fluid and selected oocytes were grouped in 50 mL conical tubes kept at 37 °C incubation until oocyte selection. Using a stereomicroscope with 80× magnification, the aspirated oocytes were transferred to a 90 mm plate. After morphological selection, the oocytes were transferred to a BIOK WASH medium twice and then moved to IVM plates. Traditional methods, as described in [22,23], for the morphological evaluation of oocyte quality are based on scoring systems and use the classification of cumulus–oocyte complexes (COCs), PBS, and spindles. COCs collected from follicles are usually classified according to the compactness of their cumulus investment and ooplasm characteristics.

#### 4.7.2. IVM

Five microdrops (90 µL) of BIOK IVM medium were distributed in 35 mm plates, with all drops covered by autoclaved mineral oil (Irvine Scientific Co. 9305). Plates were pre-equilibrated in a 5.5 ± 0.5% CO_2_ and 38.5 ± 1.0 °C atmosphere for at least 2 h before transferring the previously selected and washed oocytes. A maximum of 25 oocytes per drop, adding 10 µL of washed aspirated oocytes per drop, were kept in maturation for 24 to 26 h under a 5.5 ± 0.5% CO_2_ and 38.5 ± 1.0 °C atmosphere.

#### 4.7.3. IVF

Five microdrops (90 µL) of BIOK IVF medium were distributed in 35 mm plates, with all drops covered by autoclaved mineral oil (Irvine Scientific Co. 9305). Plates were pre-equilibrated in a 5.5 ± 0.5% CO_2_ and 38.5 ± 1.0 °C atmosphere for at least 2 h before transferring the maturated oocytes. The IVM oocytes were transferred from IVM plates, washing the oocytes twice in a BIOK TL semen medium and twice in a BIOK IVF medium. Next, we transferred the washed oocytes to the IVF plates that were previously prepared (up to 25 oocytes in each drop) and kept the IVF plates in an incubator until the moment of fertilization, while processing the semen. Semen preparation consisted of using a Percoll short gradient to recover the viable sperm in a sterile conical bottom microtube, adding 135 μL of BIOK TL medium to 135 μL of 90% BIOK Percoll, and homogenizing the solution (creating 45% Percoll). Next, we slowly deposited 260 μL of 90% Percoll on the bottom of the microtube under the 45% Percoll, and immediately after thawing the semen, (37 °C 20 s in a water bath), deposited the mixture in the microtube on the surface of the Percoll gradient. Next, we centrifuged the mixture for 7 min at 2700 G, thereby removing the supernatant, and added 1 mL of IVF medium to the formed pellet. We centrifuged the mixture again for 5 min at 950 G and collected 40 μL of the pellet that developed, which was added to the 40 μL BIOK IVF medium in another microtube. We then performed semen evaluation (motility and vigor) and corrected for the concentration of sperm/drop. A minimum 60% motility of viable sperm at a 10 × 10^6^/mL sperm concentration was used in all experiments, and 8 to 10 µL of processed semen, diluted in the BIOK IVF medium, was transferred to previously prepared IVF drops containing matured and washed oocytes. The IVF plates were kept under incubation for 16 to 18 h in a 5.5 ± 0.5% CO_2_ and 38.5 ± 1.0 °C atmosphere.

#### 4.7.4. IVC

Five microdrops (90 µL) of BIOK SOF medium were distributed in 35 mm plates, with all drops covered by autoclaved mineral oil (Irvine Scientific Co. 9305). Plates were pre-equilibrated in a 5.5 ± 0.5% CO_2_ and 38.5 ± 1.0 °C atmosphere for at least 2 h before transferring the zygotes. The zygotes were transferred from IVF plates and then washed twice in a BIOK DENUD medium and twice in a BIOK SOF medium using repeated pipetting to remove the granulosa cells, both previously equilibrated in a 5.5 ± 0.5% CO_2_ and 38.5 ± 1.0 °C atmosphere for at least 2 h before. Next, we transferred the denuded zygotes to the IVC plates previously prepared and equilibrated (up to 25 oocytes in each drop) and kept the IVC plates in an incubator for eight days. On Days 3 and 5, a half-volume of BIOK SOF medium volume (50 µL) in each drop was substituted for the same volume of pre-equilibrated new SOF medium (the same batch medium was always used). On Day 2, the cleaved zygote (%CLE) percentage was determined to quantify how many zygotes had more than two cells using a stereomicroscope at 80× amplification. On Day 7, the rate of total blastocysts (%BD7) was determined considering the sum of blastocysts (%BlD7), and that of expanded (%BxD7) and hatched (%BhD7) blastocysts was determined considering the morphological characteristics of each step of development. On Day 8, the percentage of hatched blastocysts (%BhD8) was determined considering morphological characteristics and hatched embryos. Morphological characteristics were evaluated according to Bó and Mapletof (2013) [24]. For the statistical analysis, we counted only embryos classified as degrees one and two.

### 4.8. Statistical Analysis

We established a minimum of 800 aspirated oocytes per experiment, with 200 per group. Into each IVP replicate, we equally divided the amount of aspirated oocytes into the 4 treatment groups and a minimum of 3 IVP replicates per experiment until we obtained 800 aspirated oocytes. The means and standard deviation (SD) results for the percentage ratio of each embryonic development parameter and each treatment group in Experiment 1 were analyzed. Statistical analyses were performed by applying a generalized linear mixed model assuming a beta distribution for residual effects. The model included fixed effects for the amount of the rHGAL-1 as supplementation on the IVM and SOF media. The analyses were performed using the PROCGLIMMIX procedure in SAS (version 9.4) applying a logit link function. For fixed effects, *p* < 0.05 was considered significant. The results were presented as the least squares means ± SEM using the link option to obtain estimates of the predicted probabilities and their standard errors. Standard errors on the inverse linked scale were computed by the delta method. Parameters of embryo development considered for statistical analysis were CLE, BlD7, BxD7, BhD7, BD7, BlD8, BxD8, BhD8, and BD8. The same was done for Experiment 2 (SOF medium supplementation). All percentage ratios were calculated based on the number of aspired oocytes used to initiate the IVM step.

### 4.9. Immunohistochemistry

Some embryos in different stages obtained during Experiment 2 (rHGAL-1 supplementation in the SOF medium) were subjected to immunohistochemistry to detect rHGAL-1 (exogenous protein). The embryos were fixed according to the previous protocol [25]. After the culture was completed, on the eighth day, the embryos were transferred to PBS and fixed in freshly prepared methanol/DMSO (4:1) at 4 °C overnight. After that, the embryos were transferred into freshly prepared methanol/DMSO/H_2_O_2_ (4:1:1) at room temperature for 5–10 h. The embryos were stored in groups in 100% methanol at −20 °C for later usage. On the day of the assay, the steps were as follows: (i) The embryos were rehydrated in microtubes containing 200 µL 50% methanol in PBS at room temperature for 30 min; (ii) after spinning the tube for 3 min, the solution was discharged, and we added 200 µL 0.1%BSA in PBS at room temperature for 30 min; (iii) we discharged the solution and added the coating solution (PBS containing 2% BSA and 1.:100 Biotin-Linked Polyclonal Antibody to human GAL-1, Cloud Clone Corp, LAA321Hu71) overnight at 4 degrees; (iv) the coating solution was substituted with a blocking solution (PBS containing 2% BSA) for 2 h at room temperature; (v) we executed three wash steps using 100 µL drops with PBS 0.2% BSA; (vi) the embryos were transferred to 100 µL drops with PBS 0.2% BSA and 1:500 NeutrAvidin^®^ biotin-binding protein (Thermo Scientific™, A2666) for 2 h at room temperature; (vii) we executed three washing steps in 100 µL drops with PBS 0.2% BSA; (viii) we transferred the embryos individually into 2 µL of PBS, which was placed onto a glass slide; (ix) after the PBS dried, we added 20 µL of EnVision™ FLEX DAB and 20 µL Substrate Chromogen System (DAKO, Denmark) dropwise and covered the slide using a cover glass; (x) digital photos were taken using a contrast microscope (200×) to observe darkened yellow points indicating the rHGAL-1 detected in embryo cells.

### 4.10. Bioethics Committee Approval

The installation of the Sponsoring Institution was registered for reproduction and experimentation on animals with the CEUA (Pre-clinical studies—compliance with registration requirements of Tolerana^®^ as a human biopharmaceutical product under the coordination of Dr. Eríka da S. C. Morani/CEUA.RI (Issue: 01/08/2020; Revision 01), which was approved at the CEUA meeting on 01/15/2020).

## 5. Conclusions

Based on these results, the proposed amounts of rHGAL-1 (≤20 µ·mL^−1^) are safe for use in in vitro embryo production because they do not interfere with bovine IVP embryo development, considering parameters such as CLE evaluated based on D2 and Bl, Bx, and Bh evaluated based on D7 and D8 of the Culture. We also found GAL-1 immunodetection on D8 embryonic structures, and the profile of spot intensity was dependent on exogenous supplementation. Even though it was not the objective of the study, we verified that the supplementation of 2 µg·mL^−1^ significantly improved some of the parameters of embryonic development evaluated (%BlD7, %BD7, %BlD8, %BhD8, and %BD8). Considering both pieces of information, rHGAL-1 can be used in humans to improve embryonic development for assisted reproduction.

## 6. Patent

The company: assisted by specialized lawyers, has already patented this innovation in several countries due to its highly innovative and unique technological content. This patent involved co-participation with the University of São Paulo (FFRP), world patent WO/2012/083396.

## Figures and Tables

**Figure 1 molecules-28-00859-f001:**
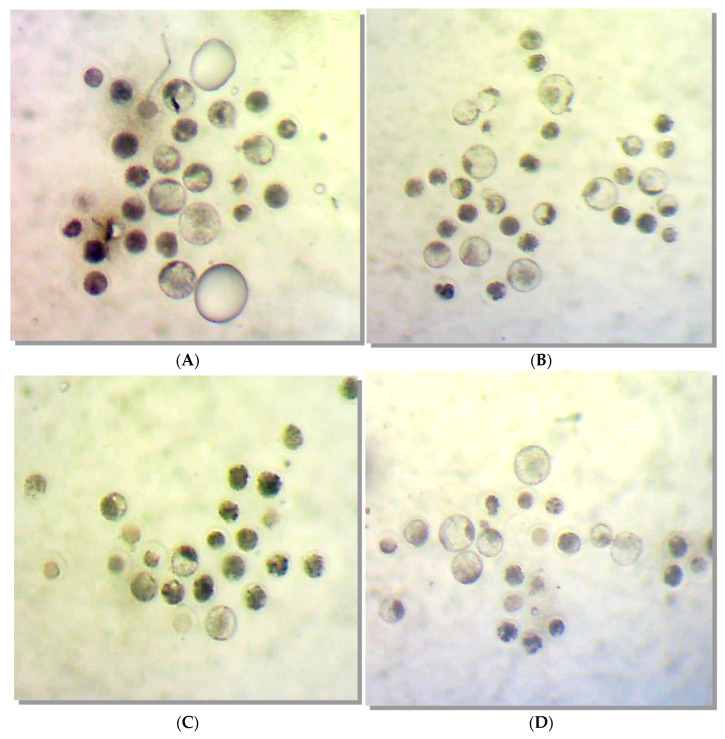
Illustrations of the in vitro bovine embryos developed in each one of the cultivated drops (Experiment 1/IVP Replicate 1.3) for each treatment group: (**A**) Group 1, (**B**) Group 2, (**C**) Group 3, and (**D**) Group 4. The size difference observed between the structures is due to the enlargement/reduction of the photo, depending on the number of embryonic structures present in each photo. The objective here is to illustrate the morphological characteristics and degrees of the embryos obtained.

**Figure 2 molecules-28-00859-f002:**
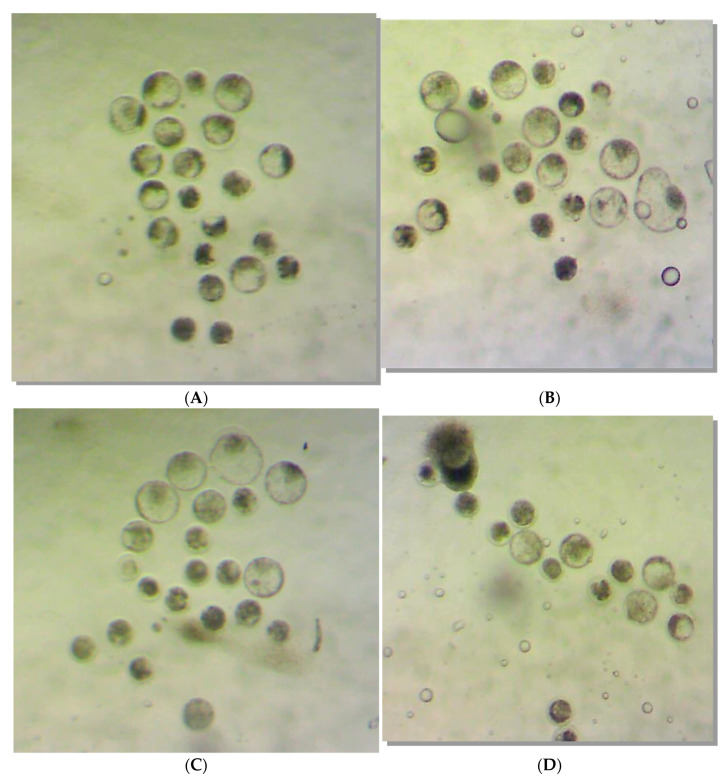
Illustrations of the in vitro bovine embryos developed in each one of the cultivated drops (Experiment 2/IVP Replicate 2.1) for each treatment group: (**A**) Group 1, (**B**) Group 2, (**C**) Group 3, and (**D**) Group 4. The size difference observed between the structures is due to the enlargement/reduction of the photo depending on the number of embryonic structures present in each photo. The objective here is to illustrate the morphological characteristics and degrees of the embryos obtained.

**Figure 3 molecules-28-00859-f003:**
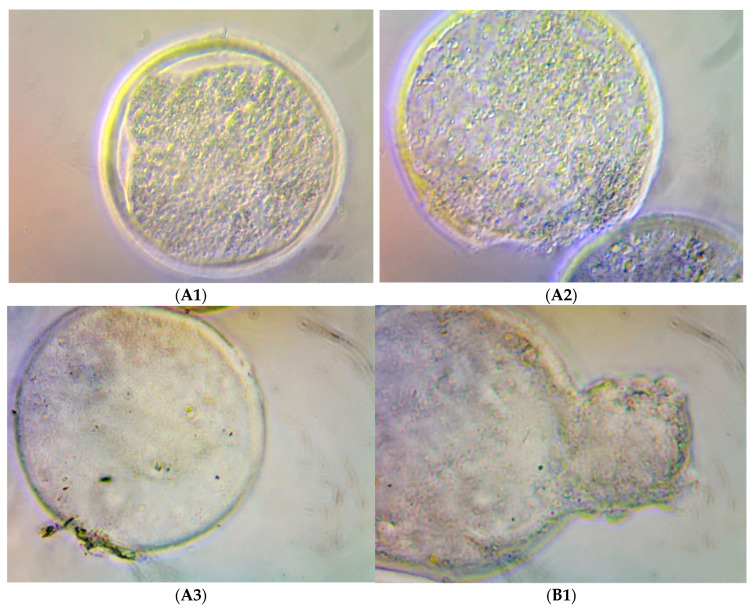
Illustrations from the contrast microscope at 200× magnification: (**A1**,**A2**) blastocysts from Experiment 1; (**A3**) expanded blastocysts from Experiment 2; (**B1**) hatched blastocyst from Experiment 2. All DAB-stained in vitro bovine embryos were cultivated without rHGAL-1 supplementation (control group).

**Figure 4 molecules-28-00859-f004:**
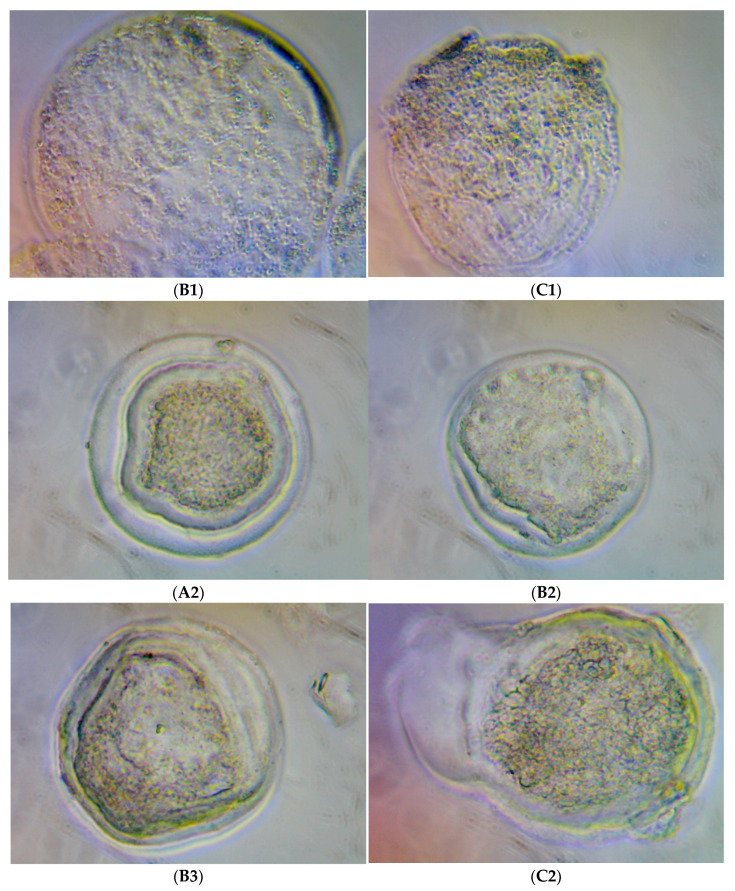
Illustrations from the contrast microscope at 200× magnification: (**B1**) expanded blastocyst from Experiment 1; (**C1**) hatched blastocyst from Experiment 1; (**A2**) blastocyst from Experiment 2; (**B2**,**B3**) expanded blastocysts from Experiment 2; (**C2**) hatched blastocyst from Experiment 2; all DAB-stained in vitro bovine embryos were cultivated with 2 µg/mL of rHGAL-1 supplementation (Group 2).

**Figure 5 molecules-28-00859-f005:**
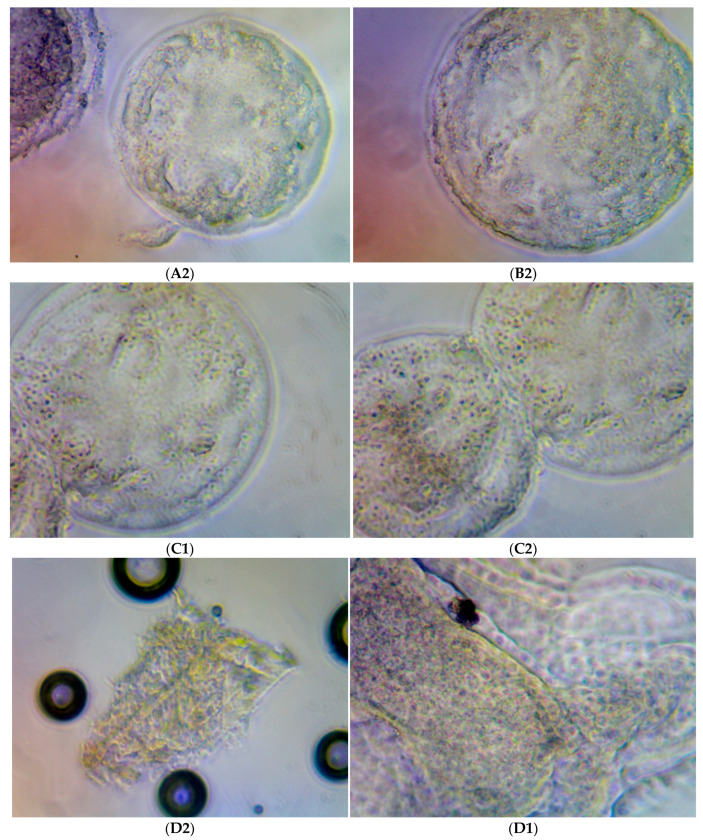
Illustrations from the contrast microscope at 200× magnification: (**A2**) blastocyst; (**B2**) expanded blastocyst; (**C1**,**C2**) hatched blastocysts; (**D1**,**D2**) trophoblast of a hatched blastocyst (bent structure when fixed to the blade). All DAB-stained in vitro bovine embryos (Experiment 2/IVP Replicate 3) used 20 µg/mL of rHGAL-1 supplementation (Group 3).

**Figure 6 molecules-28-00859-f006:**
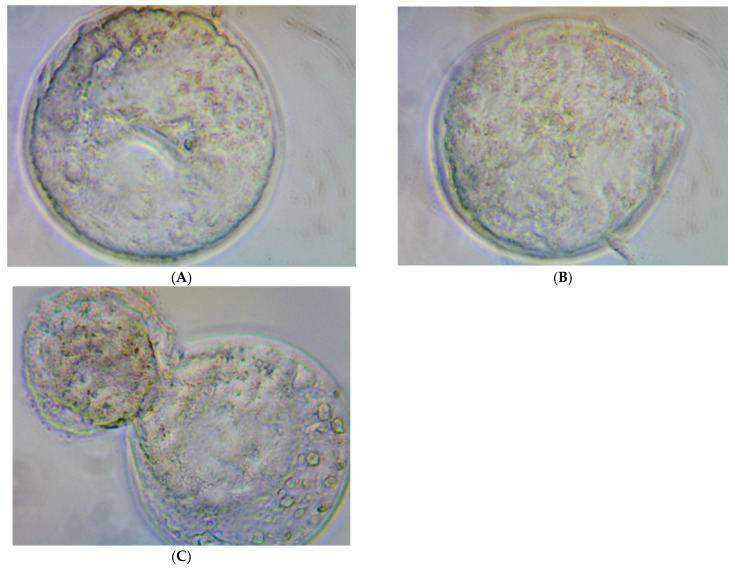
Illustrations from the contrast microscope at 200× magnification: (**A**) blastocysts; (**B**) expanded blastocyst; (**C**) hatched blastocysts; all DAB-stained in vitro bovine embryos (Experiment 2/IVP Replicate 3) used 40 µg/mL of rHGAL-1 supplementation (Group 4).

**Table 1 molecules-28-00859-t001:** Numbers (n) and percentage ratios of embryonic development parameters (CLED2; BlD7; BxD7, BhD7, BD7, BlD8, BxD8, BhD8, and BD8. All percentage ratios were calculated using the IVM oocyte number for each IVP replicate. CLED2 was evaluated 36 to 48 h after IVF and after 7 (D7) and 8 days (D8) of IVM cultivation to compare the 4 treatment groups (G1, G2, G3, and G4) and 3 IVP replicates (1, 2, and 3) used in Experiment 1 (IVM medium supplementation with test items at different concentrations for each treatment group).

Group	IVP Batch #	IVM Oocyte Numbers	CLE (D2)		BlD7		BxD7		BhD7		BD7		BlD8		BxD8		BhD8		BD8	
			$\bar{X}$	SD	$\bar{X}$	SD	$\bar{X}$	SD	$\bar{X}$	SD	$\bar{X}$	SD	$\bar{X}$	SD	$\bar{X}$	SD	$\bar{X}$	SD	$\bar{X}$	SD
G1	1.1	40	73.88 ^a^	1.01	15.19 ^ab^	0.28	10.72 ^a^	0.25	1.46 ^a^	0.09	26.37 ^ab^	0.49	7.51 ^ab^	0.13	13.67 ^a^	0.29	7.78 ^ac^	0.2	29.21 ^ab^	0.52
	1.2	110																		
	1.3	88																		
G2	1.1	33	82.02 ^a^	0.83	20.32 ^a^	0.32	11.17 ^a^	0.25	3,57 ^a^	0.1	34.07 ^a^	0.53	10.48 ^a^	0.15	11.54 ^a^	0.27	15.25 ^b^	0.27	37.69 ^a^	0.56
	1.2	116																		
	1.3	82																		
G3	1.1	34	70.69 ^a^	1.06	12.97 ^ab^	0.26	6.48 ^a^	0.23	na	na	10.30 ^ab^	0.4	6.5 ^b^	0.12	10.18 ^a^	0.25	6.66 ^ac^	0.18	23.28 ^b^	0.48
	1.2	127																		
	1.3	84																		
G4	1.1	34	79.43 ^a^	0.89	11.40 ^b^	0.25	10.53 ^a^	0.3	1,99 ^a^	0.11	17.42 ^b^	0.4	6.3 ^b^	0.14	9.65 ^a^	0.25	10.68 ^ab^	0.23	23.81 ^b^	0.48
	1.2	127																		
	1.3	96																		
Total		967																		

Different letters (a, b, c) when compared between groups (in the column) indicate statistical differences (*p* < 0.1).

**Table 2 molecules-28-00859-t002:** Numbers (n) and percentage ratios for the percentages of embryonic development parameters (CLED2; BlD7; BxD7, BhD7, BD7, BlD8, BxD8, BhD8, and BD8). All percentage ratios were calculated using the IVM oocyte number for each IVP replicate. CLED2 was evaluated 36 to 48 h after IVF and 7 (D7) and 8 days (D8) after cultivating IVM when comparing the 4 treatment groups (G1, G2, G3, and G4) and 3 IVP batches (1, 2, and 3) used in Experiment 2 (SOF medium supplementation with test items at different concentrations for each treatment group).

Group	IVP Batch #	IVM Oocyte Numbers	CLED2		BlD7		BxD7		BhD7		BD7		BlD8		BxD8		BhD8		BD8	
			$\bar{X}$	SD	$\bar{X}$	SD	$\bar{X}$	SD	$\bar{X}$	SD	$\bar{X}$	SD	$\bar{X}$	SD	$\bar{X}$	SD	$\bar{X}$	SD	$\bar{X}$	SD
G1	2.1	102	85.04 ^a^	1.99	20.90 ^ab^	3.27	12.98 ^a^	0.25	NA	NA	33.53 ^a^	2.59	6.54 ^a^	1.72	20.74 ^a^	1.45	10.01 ^a^	1.2	36.73 ^a^	0.52
	2.2	103																		
	2.3	95																		
G2	2.1	107	81.42 ^ab^	2.17	23.31 ^ab^	3.411	11.72 ^a^	0.25	NA	NA	34.67 ^a^	2.61	4.57 ^a^	1.43	19.13 ^a^	1.4	13.60 ^b^	1.37	36.62 ^a^	0.56
	2.2	104																		
	2.3	94																		
G3	2.1	105	78.21 ^bc^	2.31	27.40 ^b^	4.41	5.51 ^b^	0.23	NA	NA	32.20 ^ab^	3.13	5.06 ^a^	1.84	17.81 ^a^	1.67	11.23 ^ab^	1.54	34.74 ^a^	0.48
	2.2	104																		
	2.3	93																		
G4	2.1	110	72.64 ^c^	2.5	16.22 ^ac^	2.95	4.15 ^b^	0.3	NA	NA	22.56 ^b^	2.28	5.60 ^a^	1.94	10.74 ^b^	1.1	8.77 ^a^	1.12	24.14 ^b^	0.48
	2.2	105																		
	2.3	91																		
Total		1213																		

Different letters (a, b, c), when compared between groups (in the column) indicate statistical differences (*p* < 0.1).

## Data Availability

The data presented in this study are available on request from the corresponding author. The data are not publicly available due to privacy issues.

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
