# Peer review of "Galectin-1 Used in Assisted Reproduction—Embryo Safety and Toxicology Studies"

_molecules, 2023, doi:10.3390/molecules28020859_

Round 1
Reviewer 1 Report
The paper is important containing original data, deserves to be published, just minor spell check is required.
Author Response
Dear Reviewer
our answers are in red.
The paper is important containing original data, deserves to be published, just minor spell check is required. English language and style are fine/minor spell check require – we paid for a English Editing (attached English-Editing-Certificate-57992.pdf).
We kept the alteration controlled yet to facilitate to see the corrections done, considering the suggestions of the three reviewers.
Reviewer 2 Report
In the study, the possible effects of Galectin-1, which is used in assisted reproductive techniques, on embryo safety and toxicology were examined. The subject of this study is suitable for the “Molecules” journal. The authors indicated that using ≤20μ.mL-1 of rHGAL-1 for in vitro embryonic development makes it more reliable and safer for assisted reproduction in humans. This is an exciting study, and I suggest a few corrections to increase the scientific value of the manuscript. After these corrections are addressed by the authors, the manuscript can be accepted.
In summary, it is stated that rHGAL-1 supplements were used in 4 different concentrations. This should be changed by three different concentrations (2, 20, and 40μg.mL-1, respectively). Control group should not be included.
The last part of the summary should clearly state which concentration of rHGAL-1 was more effective.
In the last part of the introduction, the aim of the study should be clearly stated.
All paragraphs in the Materials and Methods section must begin with a capital letter; for example, “cows sent for slaughter…..” and “all procedures for in vitro bovine……..”
The results have been well presented, and the results have been supported by the discussion, but the conclusion part should be improved, especially on the fact that rHGAL-1 can be used in humans to improve embryonic development for assisted reproduction.
I think that adding the articles that I have given below will increase the quality of the article, especially IVM, IVF, and IVC.
Sen, U., Åžirin E, Onder, H., Özyürek S, Kolenda M, Sitkowska B. 2022. Macromolecules Influence Cellular Competence and Expression Level of IGFs Genes in Bovine Oocytes In Vitro. Animals, 12(19), 2604.
Sen, U., Kuran, M. 2018. Low incubation temperature successfully supports the in vitro bovine oocyte maturation and subsequent development of embryos. Asian-Australas J Anim Sci, 31(6): 827-834.
Author Response
Thank you for your comments and suggestions
our answers are in red bellow.
In the study, the possible effects of Galectin-1, which is used in assisted reproductive techniques, on embryo safety and toxicology were examined. The subject of this study is suitable for the “Molecules” journal. The authors indicated that using ≤20μ.mL-1 of rHGAL-1 for in vitro embryonic development makes it more reliable and safer for assisted reproduction in humans. This is an exciting study, and I suggest a few corrections to increase the scientific value of the manuscript. After these corrections are addressed by the authors, the manuscript can be accepted. Thank you for your comments and we’ll appreciated them, and corrections suggested.
We paid for a English Editing (attached English-Editing-Certificate-57992.pdf). We kept the alteration controlled yet to facilitate to see the corrections done, considering the suggestions of the three reviewers.
Comments:
In summary, it is stated that rHGAL-1 supplements were used in 4 different concentrations. This should be changed by three different concentrations (2, 20, and 40μg.mL-1, respectively). Control group should not be included. - done
The last part of the summary should clearly state which concentration of rHGAL-1 was more effective. – even if not the objective of the study we included this information. Concern about the limit of the words on the abstract extrapolated)
In the last part of the introduction, the aim of the study should be clearly stated – adjusted including more information.
All paragraphs in the Materials and Methods section must begin with a capital letter; for example, “cows sent for slaughter…..” and “all procedures for in vitro bovine……..” - done
The results have been well presented, and the results have been supported by the discussion, but the conclusion part should be improved, especially on the fact that rHGAL-1 can be used in humans to improve embryonic development for assisted reproduction - done
I think that adding the articles that I have given below will increase the quality of the article, especially IVM, IVF, and IVC – thank you very much for indication
Sen, U., Åžirin E, Onder, H., Özyürek S, Kolenda M, Sitkowska B. 2022. Macromolecules Influence Cellular Competence and Expression Level of IGFs Genes in Bovine Oocytes In Vitro. Animals, 12(19), 2604. https://doi.org/10.3390/ani12192604 – included in the discussion. Thank you for the indication!
Sen, U., Kuran, M. 2018. Low incubation temperature successfully supports the in vitro bovine oocyte maturation and subsequent development of embryos. Asian-Australas J Anim Sci, 31(6): 827-834 – not included
Reviewer 3 Report
It is an interesting article, but need some improvements before considering for publication.
Comments:
Please use 'replicate' instead of 'batch' throughout the text
What does exactly mean ‘tax’ please try to reformulate, use another word (percentage/ratio?) because it is hard to understand
Line 179: please change: breed, age or nutrition were not recorded/controlled
Line 208: please change 'cystic' to 'bigger than' (whether it was a cyst or not, has not been investigated, therefore should not be stated)
Line 219, 226, 248: is the spelling '05' correct?
Please add Experimental Design paragraph to summarize step by step the experiments
Results: please make it more succinct, describe only the significant differences between groups. Consider lines: 491 – 513 ?can be moved to results section?
Lines 549 – 564: this information, although interesting, is not connected with the performed study, please remove
Line 577: is that data related to this experiment? Why it 'can' be cited? Its unclear and not connected with the presented results.
Please improve Discussion – try to discuss obtained results with already published papers
Author Response
Our answer in red bellow.
It is an interesting article but need some improvements before considering for publication. Thank you for the comment
We paid for a English Editing (attached English-Editing-Certificate-57992.pdf). We kept the alteration controlled yet to facilitate to see the corrections done. We consider the three reviewers suggestions on this.
Comments:
Please use 'replicate' instead of 'batch' throughout the text - done
What does exactly mean ‘tax’ please try to reformulate, use another word (percentage/ratio?) because it is hard to understand – done altering to percentage ratio
Line 179: please change: breed, age or nutrition were not recorded/controlled - done
Line 208: please change 'cystic' to 'bigger than' (whether it was a cyst or not, has not been investigated, therefore should not be stated) - done
Line 219, 226, 248: is the spelling '05' correct? Yes …is correct and done according to the medium suppliers and routine of the lab used
Please add Experimental Design paragraph to summarize step by step the experiments – done, included item 2.2
Results: please make it more succinct, describe only the significant differences between groups. Consider lines: 491 – 513 ?can be moved to results section? These figures were included to demonstrate the differences by Immunohistochemistry assay in different embryos stages or amount of protein supplemented. They are on the results section, as cited at line 484, and please kept them on the article.
Lines 549 – 564: this information, although interesting, is not connected with the performed study, please remove – do not agree to remove. The goal of this article was demonstrated the safety of the tested item for reproduction assisted procedures. We demonstrate that using doses < 20ug/mL the item is safety. However, as a complemented conclusion this study demonstrates a possibility of rhGAL-1 supplementation on IVP mediums to improve the embryo development in vitro. The comments cited on those lines describing a resume of the status of the art of GAL-1 used to improve the pregnancy rate when administrated into the uterus (local administration). We complement the initial of the paragraph to try to get better interpretation. Let me know if this is enough.
Line 577: is that data related to this experiment? Why it 'can' be cited? Its unclear and not connected with the presented results. We tried to alter the text to get better interpretation. If the question is the indication of Gal supplementation of IVP mediums to get improve of in vitro embryo development, yes agree that is not the goal of this Study and was cited as a suggestion and need more studies. This is match with the comments of the other reviewers, included. Let me know f this is enough.
Please improve Discussion – try to discuss obtained results with already published papers. Some alterations are done. Remembering that rHGAL-1 is never used as a kind of supplementation by medium. The goal of this article was demonstrating the security amount of protein that is reasonable to use in human assisted reproduction procedures with pharmacological safety for the embryos. For the mother, another nonclinical trials were done. And it is safety when used < 5mg.mL for mother.